# Enrichment of Sunflower Oil with Ultrasound-Assisted Extracted Bioactive Compounds from *Crithmum maritimum* L.

**DOI:** 10.3390/foods11030439

**Published:** 2022-02-02

**Authors:** Gabriela Sousa, Mariana I. Alves, Marta Neves, Carla Tecelão, Suzana Ferreira-Dias

**Affiliations:** 1Instituto Superior de Agronomia, Universidade de Lisboa, LEAF, Linking Landscape, Environment, Agriculture and Food, Associated Laboratory TERRA, 1349-017 Lisbon, Portugal; gabriela.spsousa@gmail.com (G.S.); marianaialves98@gmail.com (M.I.A.); 2MARE-Marine and Environmental Sciences Centre, ESTM, Politécnico de Leiria, 2520-641 Peniche, Portugal; marta.neves@ipleiria.pt (M.N.); carla.tecelao@ipleiria.pt (C.T.)

**Keywords:** antioxidant activity, flavonoids, halophyte, optimization, phenolic compounds, pigments, response surface methodology, sea fennel

## Abstract

*Crithmum maritimum* L., or sea fennel, is an edible halophyte plant, rich in phenolic compounds with antioxidant and antimicrobial activities, that naturally grows in Mediterranean coasts. This study aims to incorporate bioactive compounds extracted from lyophilized *Crithmum maritimum* to sunflower oil assisted by ultrasounds (UAE), to improve its biological value and oxidative stability. UAE conditions were optimized as a function of time (5–20 min) and lyophilized plant concentration (5–20% *m*/*v*). The experiments were dictated by a central composite rotatable matrix. Oxidation products were not influenced by UAE conditions. Acidity, chlorophyll, and carotenoid contents were affected by both factors, while total phenols, flavonoids, and antioxidant activity (FRAP method) only increased with plant concentration. Response surfaces were fitted to these experimental results. Flavonoids were highly related with oil antioxidant activity. No sensory defects were detected in supplemented oil (12.5% *m*/*v* plant/5 min UAE). The oxidative stability of this oil was evaluated at 60 °C/12 days. Chlorophylls, phenols, radical scavenging (DPPH), and antioxidant activities decreased over time but were always higher than the values in non-supplemented oil (8.6 and 7-fold with FRAP and DPPH, respectively). *C. maritimum* presented high amounts of bioactive compounds with antioxidant activity, adequate for sunflower oil supplementation by UAE.

## 1. Introduction

Lipid oxidation is a big constraint for the food industry, causing a decrease in nutritional value and shelf life of oils and food products. To overcome this problem, the food industry has been using synthetic antioxidants, namely butylated hydroxyanisole (BHA), butylated hydroxytoluene (BHT), and tertiary butyl hydroquinone (TBHQ) [1]. However, due to safety issues, the number of allowed synthetic antioxidants that can be incorporated into food products has been decreasing [2]. Consequently, the search for natural antioxidants (mostly from plant origin) to be added to edible oils has been an important issue for the food industry [3]. In this context, halophyte plants have recently drawn some attention in the food markets worldwide. These plants can tolerate extreme environmental conditions, particularly high levels of salinity, enabling them to grow in coastal areas and in arid and semiarid regions. In a broad sense, halophytes are recognized as a valuable source of minerals and fibers, and some species may also provide important contents of protein and high-quality lipids [4]. Halophytes synthesize various metabolites, namely phenolic compounds that can offer multiple benefits in the food, cosmetics, and pharmaceutical industries due to their antioxidant and antimicrobial activities [4,5]. Flavonoids are phenolic substances widespread in plants that exhibit antioxidant activity. Thus, flavonoids can offer benefits in various products, such as food products [6]. Additionally, halophyte plants could represent a good alternative to other agroforest raw materials because they can grow in the presence of low-quality saline soils, which are not adequate for conventional agriculture [7,8].

*Crithmum maritimum* L. (sea fennel) is an edible and medicinal halophyte plant that naturally grows in coastal ecosystems with a Mediterranean climate [7]. This halophyte contains significant amounts of valuable compounds such as ascorbic acid, carotenoids, tannins, flavonoids, and other polyphenolic compounds, that show antioxidant and antimicrobial activities [8]. Hence, *C. maritimum* is a promising wild edible plant that can be applied to improve and preserve food products [9].

Ultrasound-assisted extraction (UAE) of bioactive compounds directly from the lyophilized plant into the oil can be performed. This technique offers several advantages in comparison with conventional extraction procedures, namely (i) the use of organic solvents is avoided or minimized; (ii) low temperature and extraction time are required, leading to less energy consumption; (iii) high extraction yields are achieved, and (iv) the extract quality is preserved [10,11]. UAE has been successfully employed in several food matrices (e.g., fruits, vegetables, and edible oils) for the extraction of bioactive compounds, such as polyphenols, carotenoids, and polysaccharides [10,11,12,13].

In this sense, the purpose of this study was to supplement sunflower oil with bioactive compounds from *C. maritimum* L., directly extracted from the plant to the oil by UAE, aiming at increasing the nutritional and economic values and oxidative stability of the oil. The UAE process was performed under the conditions dictated by a central composite rotatable design (CCRD) as a function of lyophilized plant/oil ratio and ultrasound application time, to find the optimized conditions. The storage stability of sunflower oil enriched with bioactive compounds from *C. maritimum*, under selected UAE conditions, was additionally assessed at 60 °C.

## 2. Materials and Methods

### 2.1. Materials

Refined sunflower oil without added antioxidant compounds was kindly donated by SOVENA Portugal, Consumer Goods SA. The halophyte plant *Crithmum maritimum* L. was collected in Peniche, Portugal (39°21′53.1″ N, 9°24′15.8″ W) in February and March 2021. Gallic acid (97.5% purity), quercetin (95% purity) and Trolox (97% purity) used for calibration curves were purchased from Sigma and Acros Organics; β-carotene (>97% purity) was purchased from TCI Europe. All reagents and solvents used were *p.a.* and obtained from different sources.

### 2.2. Characterization of Crithmum maritimum

The halophyte (whole plant) was lyophilized at −56 °C for 3 days under vacuum (freeze-dryer Telstar, Lyoquest-85, Telstar Portugal, Lisbon, Portugal) and characterized concerning (i) pigments-chlorophyll and carotenoid contents, (ii) total phenolic compounds and flavonoids, and (iii) antioxidant activity. The lyophilized *C. maritimum* was milled in a knife mill (Retsch SM 2000; Retsch GmbH, Haan, Germany) with an exit grid of 1 mm and stored at 20 °C in the dark until use. Methanolic and *n*-hexane extracts of the lyophilized plant (5% *m*/*v*) were prepared by vortex stirring the biomass in the solvent for 6 min. Then, the suspensions were centrifuged (10 min at 4000 rpm; Hermle Z 383 K, Wehingen, Germany) to recover the liquid phase. The obtained extracts were stored at 4 °C protected from light, until subsequent analyses. Except for chlorophyll assay, the same methodologies, previously used by us for the characterization of the macroalgae *Pelvetia canaliculata*, were followed [14].

#### 2.2.1. Contents of Chlorophylls and Carotenoids of *C. maritimum*

The content of chlorophylls of *C. maritimum*, expressed as mg/kg d.w., was determined according to Lichtenthaler and Buschmann [15] as briefly described. Samples were freeze-dried and 0.2 g were homogenized with acetone using two volumes of 5 mL, under stirring in a vortex mixer, for 5 min. After 10 min centrifugation (13,000× *g*), the supernatant was filtered and recovered in a 10 mL volumetric flask. The final volume was completed with acetone. The absorbance of the extract was read at 644.8 and 661.6 nm (Thermo Scientific, Evolution 201 Spectrophotometer) using acetone as reference.

The content of carotenoids of the halophyte plant was assayed in the *n*-hexane extract after the dilution in *n*-hexane (1:1 ratio v/v), according to Rougereau et al. [16] as previously described [14]. A calibration curve was prepared with β-carotene in concentrations up to 19.1 µg/mL in *n*-hexane. The absorbance of the standards and samples was read at 450 nm using *n*-hexane as reference. The results were expressed as mg β-carotene/kg lyophilized plant (d.w.). The obtained calibration curve presented a determination coefficient, R^2^, of 0.995. Three replicates of each sample were analyzed.

#### 2.2.2. Contents of Total Phenolic Compounds and Flavonoids of *C. maritimum*

Total phenolic content of *C. maritimum* methanolic extract was determined in a UV-Vis spectrophotometer (Agilent Technologies Cary series 100 UV-Vis, Santa Clara, CA, USA), at 765 nm using the Folin–Ciocalteu method, as described by Matanjun et al. [17]. The results were expressed as mg gallic acid/kg *C. maritimum*. A calibration curve for phenolics, using gallic acid in concentrations up to 0.5 mg/m, was prepared. The obtained curve had a determination coefficient, R^2^, of 0.988.

Flavonoid content was spectrophotometrically determined at 510 nm according to the method described by Laulloo et al. [18] and used by Sousa et al. [14] for *Pelvetia canaliculata*. The results were expressed in mg quercetin equivalent/kg of plant (d.w.). A calibration curve was prepared using quercetin in concentrations up to 1.2 mg/mL (R^2^ = 0.993). Samples were analyzed in triplicate.

#### 2.2.3. Assessment of the Antioxidant Activity of *C. maritimum*

##### DPPH Radical Scavenging Activity

Radical scavenging activity of the methanolic solution of *C. maritimum* was determined using the DPPH radical scavenging assay, as described by Brand-Williams et al. [19]. The results were expressed as % Radical Scavenging Activity (% RSA). A calibration curve was prepared with the synthetic antioxidant Trolox (6-hydroxy-2,5,7,8-tetramethylchroman-2-carboxylic acid) in concentrations up to 0.25 mg/mL (R^2^ = 0.9938). The absorbance was read at 515 nm against methanol. The obtained results were also presented in equivalent of Trolox (mg/kg of plant, d.w.). Three replicates of each sample were analyzed.

##### Ferric Reducing Antioxidant Power (FRAP)

The antioxidant capacity of the *C. maritimum* methanolic extract was determined spectrophotometrically at 595 nm according to Benzie and Strain [20] with slight modifications described by Sousa et al. [14]. A calibration curve was prepared using Trolox in concentrations up to 0.25 mg/mL (R^2^ = 0.991). Trolox samples were treated following the same procedure. The obtained results were expressed in equivalent of Trolox (mg/kg of plant). The analyses were carried out in triplicate.

### 2.3. Ultrasound Assisted Extraction of Bioactive Compounds to Sunflower Oil

The bioactive compounds of *C. maritimum* were directly extracted to the oil using an ultrasound bath (Transsonic TS 540; ultrasound frequency of 35 kHz). Extraction experiments were performed under the conditions dictated by the experimental design (CCRD), as a function of two factors: UAE time and *C. maritimum* concentration. This experimental matrix is one of the designs used in Response Surface Methodology (RSM) [21]. In CCRD, each variable is tested in five levels (quantities), allowing to fit curved surfaces to the experimental data, which are described by second order polynomials. The CCRD matrix is formed by four factorial points (experiments 1 to 4), four star-points (experiments 5 to 8), and four central points (experiments 9 to 12), comprising 12 experiments. The CCRD followed was the same used with *Pelvetia canaliculata* for supplementation of sunflower oil by ultrasound-assisted extraction [14]. Therefore, UAE time varied from 5 to 20 min and the concentration of lyophilized *C. maritimum* varied from 5 to 20% *m*/*v* (Table 1). Two extra samples (samples 13 and 14) were prepared and analysed. Sample 13 corresponded to the original sunflower oil, without any treatment, while sample 14 is the sunflower oil sample submitted to 20 min ultrasounds.

The lyophilized plant was added to 40 mL of sunflower oil in amounts dictated by the CCRD. Then, the samples were stirred in a vortex mixer and submitted to UAE for 5 to 20 min. After UAE, the samples were centrifuged (10 min at 4000 rpm in a Hermle Z 383 K centrifuge, Wehingen, Germany) to recover the supplemented oil. Finally, the samples of supplemented oil were stored in the refrigerator at 4 °C protected from the light until further analyses.

### 2.4. Analyses of Sunflower Oil

#### 2.4.1. Chemical Quality Parameters of Supplemented Sunflower Oils

The amounts of free fatty acids (acidity) in sunflower oil samples were assayed by titration with 0.1 M potassium hydroxide solution. The results were presented in acid value (mg KOH/g oil). The oxidation products in sunflower oil samples were indirectly assessed by the absorbance at 232 nm, K_232_ (related with the presence of primary oxidation products, conjugated dienes) and at 270 nm, K_270_ (related with the presence of secondary oxidation products) of oil samples in isooctane (1%, *m*/*v*) as described in the Commission Regulation (EEC) No. 2568/91 [22] for olive oil and olive pomace oil. The monitorization of K_232_ and K_270_ in supplemented oils and comparison with the initial values allows the detection of eventual oxidation phenomena occurring during UAE. The peroxide value (PV) of the initial oil and of the sample obtained by supplementation with 12.5% (*m*/*v*) *C. maritimum* under the shortest UAE time (5 min) was determined, and the results were expressed in mEq O_2_/kg [22]. The PV for the other samples of the CCRD was not determined due to the lack of sufficient volumes of oil samples for all the analysis. Three replicates of each sample were analyzed.

#### 2.4.2. Contents of Chlorophylls and Carotenoids in Sunflower Oils

The chlorophyll content of initial and supplemented sunflower oil samples was spectrophotometrically assayed following the methodology described by Pokorný et al. [23]. The absorbance of the samples was directly measured in a UV-Vis double beam spectrophotometer (Agilent Technologies Cary series 100 UV-Vis, Santa Clara, CA, USA), using air as reference, at the following wavelengths: 630 nm, 670 nm, and 710 nm. The results were expressed in mg pheophytin a/kg oil. The content of carotenoids of the initial and supplemented sunflower oil samples was determined as previously described (cf. 2.2.1.) and expressed as mg of β-carotene/kg oil. Samples were analyzed in triplicate.

#### 2.4.3. Determination of Total Phenolic Compounds and Flavonoids in Sunflower Oils

The preparation of phenolic extracts was carried out according to Hrncirik et al. [24]. Phenolic and flavonoid contents were assayed according to the method described in 2.2.2. and the phenolic contents were expressed as mg of gallic acid/kg of oil and the flavonoids as mg of quercetin/kg of oil, respectively. Three replicates of each sample were analyzed.

#### 2.4.4. Determination of Antioxidant Activity of Sunflower Oils

Antioxidant activity of sunflower oil samples was assayed by DPPH and FRAP methods, as previously described (c.f. 2.2.3.). The obtained results were presented as equivalent mg of Trolox/kg of oil. The analyses were carried out in triplicate.

### 2.5. Sensory Evaluation of Supplemented Sunflower Oil

The sample of supplemented sunflower oil obtained with 12.5% (*m*/*v*) of halophyte, under 5 min of UAE, was sensory evaluated by a flavour profile panel of nine assessors trained for oil and olive oil sensory analysis [22]. Panelists were asked to detect, identify, and quantify the intensity of rancid and other eventual off-flavours, as well as positive flavours coming from the halophyte plant using a continuous and unstructured scale, from 0 (absence) to 10 (extremely strong intensity). The coded samples were sensory evaluated at 28–30 °C. The obtained results were used to calculate the mean and median of the detected attributes.

### 2.6. Oxidation Tests under Accelerated Conditions

The sunflower oil sample, obtained by supplementation with 12.5% *C. maritimum* using the shortest UAE time (5 min), as well as non-supplemented oil (control), were submitted to accelerated oxidation, following the methodology described by Sousa et al. [14]. Amber glass flasks (10 mL) were filled with these oil samples (full of oil to avoid contact with oxygen) and stored in an oven at 60 °C for 12 days, in the dark. Daily, aliquots corresponding to individual flasks were collected and analysed.

### 2.7. Statistical Analyses

The treatment of the results of each parameter evaluated for the 12 samples obtained from the CCRD was performed using the software “Statistica”, version 7, from Statsoft, Tulsa, OK, USA. Both linear and quadratic effects of each factor (variable), i.e., plant concentration or UAE time, and the interaction effect (plant concentration × UAE time) on each assayed parameter, were calculated. ANOVA was used to evaluate the significance of the effects of the factors, and determine which effects were considered significant or not. Usually, significant effects must have a *p* ≤ 0.05. However, factors with *p* > 0.05 must be retained when their removal will decrease the goodness of fit of the model. Haaland [25] and Rodrigues and Iemma [26] indicated a *p* value < 0.1 for the selection of variables to include in RSM models. A response surface described by a first- or a second order polynomial equation was fit to each set of results. The values of the determination coefficient (R^2^) and of the determination adjusted coefficient (R^2^_adj_) were used to evaluate the quality of the adjustment of the fitted models to the experimental results.

The fit of first-order decay models to the results of the decay of bioactive compounds along accelerated oxidation test, was carried out using the MS Excel “Solver” supplement. The fit was performed by minimizing the residual sum-of squares between the experimental data points and those estimated by the respective model.

## 3. Results

### 3.1. Characterization of Crithmum maritimum

Chlorophylls and carotenoids are natural pigments responsible for the colour of plants and algae. Chlorophylls are responsible for the green colour while carotenoids are responsible for orange/yellow colour. Additionally to their contribution to colour, pigments also perform important physiological functions. While chlorophylls may act as prooxidants, carotenoids have antioxidant activity. Therefore, it is important to quantify them and understand how they behave when incorporated in a food matrix [27]. 

The quantification of these pigments in *Crithmum maritimum* was carried out and the results are presented in Table 2. The analyses show that *C. maritimum* has 1092 mg of chlorophylls and 470 mg of β-carotene per kg of lyophilized plant.

Renna et al. [28] determined the chlorophyll content of *C. maritimum* collected in Italy, using an extraction procedure with 80% acetone followed by spectrophotometric quantification at 647 and 664 nm. Values of 32.71 ± 3.59 mg for chlorophyll a and 9.75 ± 0.86 mg for chlorophyll b per 100 g of freeze-dried halophyte were attained. Hence, the authors quantified a total amount of chlorophylls a and b of 424.6 mg/kg. Our sample of *C. maritimum* presented 2.6-fold the value obtained by Renna et al. [28]. This might be explained by different environmental conditions where the plant was grown. Nabet et al. [29] determined the content of carotenoids of *C. maritimum* harvested in Algeria by carrying out a double extraction with *n*-hexane/acetone/ethanol (2/1/1, *v*/*v*/*v*). After washing the mixture with distilled water, the *n*-hexane phase was recovered for spectrophotometric quantification of total carotenoids at 450 nm. The results obtained by these authors (62.6 ± 3.8 mg of β-carotene/kg of dry weight) were almost 8 times smaller than the results obtained in the present study. These large differences may be due to the use of a different method of quantification, which includes a previous extraction, and by different environmental conditions.

Table 2 also presents the total phenolic and flavonoid contents and antioxidant activity (assessed by DPPH and FRAP assays) of methanolic extracts of *C. maritimum*. These analyses demonstrated the presence of 8110 mg of gallic acid/kg of lyophilized halophyte, 56,202 mg of quercetin/kg of lyophilized halophyte, a radical scavenging activity of 87.88% (7320 mg trolox/kg lyophilized halophyte) and an antioxidant activity (assessed by FRAP) of 20,997 mg equivalent of Trolox/kg of lyophilized halophyte.

Houta et al. [30] determined the total phenolic content in different parts of *C. maritimum* plant and quantified 11.52 ± 0.06 mg of gallic acid/g d.w. in leaves and 13.73 ± 0.41 mg of gallic acid/g d.w. in stems. In our study, a mixture of leaves and stems was lyophilized and analysed. The obtained value (8.11 mg of gallic acid/g d.w.) is slightly smaller than the values reported by Houta et al. [30]. In addition, Nabet et al. [29] found a phenolic content of *C. maritimum* extracts almost six–fold the value determined in our sample (47.1 ± 0.1 g of gallic acid/kg d.w.). These differences may be explained by several factors that influence the phenolic content of halophyte plants, namely the harvest time, biotic/abiotic stressors (e.g., salinity, UV radiation, extreme temperatures, and exposure to pollutants), growth conditions (water and soil composition) and extraction methodology [4].

Moreover, Houta et al. [30] also determined the flavonoid content in different parts of the plant and obtained 3.93 ± 0.1 mg of quercetin/g d.w. in the leaves and 4.92 ± 0.3 mg of quercetin/g d.w. in stems. In the present study, the content of flavonoids in the mixture of stems and leaves of *C. maritimum* (56.20 mg of quercetin/g d.w.) is more than ten-fold the values found by Houta et al. [30] which may be explained by a different extraction process and by different environmental conditions. The extraction process carried out by Houta et al. [30] involved stirring the dry powder in methanol for 24 h which, if performed without protection from the light, may be responsible for the degradation of flavonoid compounds [31].

### 3.2. Quality Parameters of the Supplemented Sunflower Oil

The supplemented oil samples were prepared following a CCRD matrix, using different *C. maritimum* concentrations and UAE times. The obtained results, concerning the quality parameters of these oils, were used to (i) calculate the linear and/or quadratic effects of the variables and the interaction effect between plant concentration and UAE time, (ii) to fit a response surface to each set of results, and (iii) to decide about the best UAE conditions to supplement sunflower oil.

#### 3.2.1. Acidity

The acidity of an oil or a fat, due to the presence of free fatty acids, is a quality parameter related to the hydrolysis of triacylglycerols. For refined edible oils, the legal limit of free fatty acids, expressed as acid value (AV), is 0.6 mg KOH/g oil [32].

In the present study, the acid value of the supplemented sunflower oils varied between 0.30 and 0.51 mg KOH/g oil, while the non-supplemented original sunflower oil had an AV of 0.13 mg KOH/g oil Although the AV of all the samples was below the legal limit, the statistical analysis showed that both UAE time and plant load have a positive linear effect on the acidity value (*p* = 0.03 and *p* = 0.10, respectively). Additionally, the variable “plant concentration” presented a significant quadratic positive effect (*p* = 0.03) on oil acidity, showing that the AV can be described by a concave surface as a function of this variable. The importance of the quadratic effect of UAE on oil acidity indicates that this effect must be considered in the response surface model. Conversely, the interaction effect was not significant and, therefore, was to be ignored. Thus, the acidity value in the supplemented sunflower oil can be fitted to the response surface shown in Figure 1 (R^2^ = 0.762; R^2^_adj_ = 0.626), and described by the following second order polynomial Equation (1): (1)$$AV\;\left(mg\;KOH/g\;oil\right)=0.568-0.039\left[plant\right]+0.002\left[plant^{2}\right]-0.014t+0.001t^{2}$$
where [*plant*] is the concentration of lyophilized *C. maritimum* (d.w.), expressed in % (*m*/*v*), and *t* corresponds to ultrasound assisted extraction time (min).

#### 3.2.2. Oxidation Products

The oxidation stage of an oil can be indirectly evaluated spectrophotometrically at 232 nm and 270 nm. The absorbance at 232 nm is due to the presence of primary oxidation products–conjugated dienes, while the secondary oxidation products like short chain fatty acids, ketones, and aldehydes absorb at 270 nm. The original oil showed a K_232_ of 2.85 and a K_270_ of 2.31. The supplemented oils presented K_232_ values ranging from 2.75 and 3.12 and K_270_ varying from 2.16 to 2.47. In fact, the statistical analysis of these results showed that neither the extraction time nor plant concentration had significant impact on oil oxidation. Moreover, the PV of the initial oil and of the supplemented oil obtained by UAE using 12.5% plant during 5 min (trial n° 7) were 2.5 and 2.6 meq O_2_/kg, respectively. These values are far below from the legal limit (10 meq O_2_/kg) and show that UAE did not affect the oxidation stage of the oil.

### 3.3. Green Pigments and Carotenoid Contents in Sunflower Oil

The supplemented oils exhibited colours between yellow and intense green, in contrast with the very light-yellow colour of the original refined sunflower oil. Figure 2 shows a picture of the nine different supplemented sunflower oils under the conditions (lyophilized *C. maritimum* concentration and UAE time) presented in Table 3. The colour changes observed in supplemented oils indicate the extraction of pigments from the plant to the oil by the effect of ultrasounds.

Green pigments (chlorophylls) and carotenoid contents of supplemented sunflower oil samples, and their respective extraction yields (Ƴ) from *C. maritimum* to the oil by UAE, are shown in Table 3. All supplemented oils showed higher contents of green pigments and carotenoids than the original refined sunflower oil which contained 0.03 ± 0.01 mg pheophytin a/kg oil and 1.25 ± 0.66 mg β-carotene/kg oil. Extraction yields of these groups of compounds, from the plant to the oil, were calculated as previously described [14].

The content of green pigments in supplemented sunflower oil samples varied between 9.14 mg pheophytin a/kg oil and 44.24 mg pheophytin a/kg oil, while the extraction yield from *C. maritimum* to the oil during ultrasound treatment varied from 12.6% to 20.9%.

The statistical analysis of these results showed that the chlorophyll content linearly increased with plant concentration (*p* = 0.0012). The interaction effect of the two factors was not significant. However, the quadratic effects of both factors are important enough to be considered in the response surface model, even having *p* values higher than 0.05 [25,26]. Thus, the chlorophyll content in the supplemented oil can be fitted to the convex response surface shown in Figure 3 (R^2^ = 0.850; R^2^_adj_ = 0.794) and described by the following second-order polynomial Equation (2): (2)$$\left[pheophytin\;a\right]\;\left(mg/kg\;oil\right)=-11.726+4.3176\left[plant\right]-0.0938\left[plant^{2}\right]+0.006t^{2}$$
where $\left[plant\right]$ corresponds to the lyophilized plant concentration, in % (*m*/*v*), and t corresponds to ultrasound extraction time (min).

In UAE from *Pelvetia canaliculata* to sunflower oil, Sousa et al. [14], using the same operation conditions, obtained supplemented oils containing 25.84 to 72.11 mg pheophytin a/kg oil, and extraction yields from 54.7 to 88.4%. They observed that an increase in the amount of the seaweed caused an increase in chlorophyll content of the samples. However, they also observed a significant linear negative effect of UAE time and a significant quadratic negative effect of algae concentration in chlorophyll content of the supplemented sunflower oil.

Carotenoid content in supplemented sunflower samples varied between 12.63 mg β-carotene/kg oil (sample 5) and 53.47 mg β-carotene/kg oil (sample 3), and the extraction yields of carotenoids from *C. maritimum* to the oil ranged from 30.9% to 57.9% during ultrasound treatment (Table 3). Statistical analysis of the results showed that the extraction of carotenoids to the oil linearly increased with plant concentration (*p* = 0.0024). The quadratic effect of UAE time and the interaction of UAE time and plant concentration cannot be ignored in the response surface model. Their removal would cause a lack of fit of the model. Therefore, the experimental results could be fitted to the response surface in Figure 3 (R^2^ = 0.811; R^2^_adj_ = 0.741) described by the following Equation (3):(3)$$\left[\beta -carotene\right]\;\left(mg/kg\;oil\right)=-8.526+4.154\left[plant\right]+0.084t^{2}-0.175\left[plant\right]t$$
where $\left[plant\right]$ corresponds to the plant concentration, in % (*m*/*v*), and t corresponds to ultrasound extraction time (min).

The concave response surface shows that the highest carotenoid contents are obtained for higher plant concentrations and UAE times below 10–12 min.

These results are similar to those observed by Corbu et al. [33] in UAE from dried sea buckthorn by-products (DSB) to refined sunflower oil. They observed that an increase in the amount of DBS from 2.5% to 10.0% (*m*/*v*) caused an increase in the carotenoid content. Moreover, when refined sunflower oil was supplemented with *P. canaliculata* by UAE [14], the final contents of carotenoids varied from 3.87 to 8.23 mg/kg oil, which are much smaller values than those obtained in *C. maritimum* supplemented oils. In fact, the lowest content of carotenoids in *C. maritimum* supplemented oil was about 3.3-fold the minimum value for *P. canaliculata* supplemented oils, while the highest content in *C. maritimum* supplemented oil was 6.5 times the highest value achieved with *P. canaliculata* supplementation. Since carotenoids are bioactive antioxidant compounds, the results obtained with *C. maritimum* are very promising to produce oils with improved bioactivity and stability.

### 3.4. Determination of Total Phenolic Compounds, Flavonoids, and Antioxidant Activity

#### 3.4.1. Total Phenolic Compounds and Flavonoids Contents

Table 4 shows the phenolic content of supplemented sunflower oil samples, and the respective extraction yields. Since *C. maritimum* had 8110 mg gallic acid/kg of lyophilized plant, an increase in phenolic content of supplemented sunflower oil samples was expected. In fact, all the supplemented sunflower oil samples had a higher phenolic content (32.66 to 72.13 mg gallic acid/kg oil) than the non-supplemented samples (10.14 and 4.12 mg gallic acid/kg oil). Nevertheless, the extraction yields from the plant to the oils were low, ranging from 2.8% to 6.8%, which may be ascribed to the low solubility of the phenolic compounds of *C. maritimum* in refined sunflower oil. The solubility of phenolic compounds depends on the raw material, the extraction method, their degree of polymerization, the interaction with the other constitutes and the type and the effect of the solvent used [34].

The extraction yield of the phenolic compounds from *P. canaliculata* to refined sunflower oil [14] were even lower (0.92–3.10%) than the yield obtained with *C. maritimum* under similar UAE conditions. Moreover, the phenolic contents of *Pelvetia* supplemented oils (8.68–26.41 equivalent mg of gallic acid/kg oil) were much lower than the values obtained in the present study in *C. maritimum* supplemented oils.

The statistical analysis of the results only showed a significant linear positive effect of plant concentration (*p* = 0.025) and the effect of the quadratic term of UAE time equal to 0.068. Consequently, the amount of phenolics in supplemented sunflower oils could be fitted to the response surface in Figure 4 (R^2^ = 0.696; R^2^_adj_ = 0.629), described by the following Equation (4):(4)$$\left[gallic\;acid\right]\left(mg/kg\;oil\right)=26.618+14.879\left[plant\right]+0.0043t^{2}$$
where $\left[plant\right]$ corresponds to the lyophilized plant concentration, in % (*m*/*v*), and *t* is the UAE time in min.

Sousa et al. [14] studied the phenolic content of sunflower oil supplemented with different amounts of *Pelvetia canaliculata*, extracted by ultrasounds and concluded that seaweed concentration had a significant negative linear effect on phenolic content, the opposite of what was verified in the present study. However, they also concluded that seaweed concentration had a significant positive quadratic effect.

The flavonoid content of supplemented sunflower oil samples, as well as the extraction yields, are shown in Table 4. Since *C. maritimum* had 56,202 mg quercetin/kg of lyophilized plant, an increase in the flavonoid content of supplemented sunflower oil samples was expected, which was observed. However, the extraction yields from the plant to the oils were rather low, ranging from 1.9% to 3.4%. Again, as occurred with total phenolic compounds, it may be explained by the low solubility of the flavonoid compounds of *C. maritimum* in the oil. The extraction of plant flavonoids is influenced by the solvent polarity [35] that in our case is the sunflower oil which acts as a non-polar solvent.

Statistical analysis of the flavonoid content in supplemented oils only showed a significant linear positive effect of lyophilized plant concentration (*p* = 0.002). Thus, the flavonoid content could be fitted to the response surface shown in Figure 4 (R^2^ = 0.843; R^2^_adj_ = 0.825) described by the following Equation (5):(5)$$\left[quercetin\right]\;\left(mg/kg\;oil\right)=36.63+14.71\left[plant\right]$$
where $\left[plant\right]$ corresponds to the concentration of lyophilized *C. maritimum*, in % (*m*/*v*).

Conversely, Sousa et al. [14] did not observe any relationship between *Pelvetia canaliculata* concentration and flavonoid content of supplemented oils.

#### 3.4.2. Antioxidant Activity of Supplemented Sunflower Oil Samples

The antioxidant activity of initial oil and supplemented oils was evaluated by the assay of the radical scavenging activity determined by the DPPH methodology, as well as by the antioxidant activity expressed in mg Trolox equivalent/kg oil, assayed either by DPPH or by FRAP methods. These results are presented in Table 4.

The methanolic extract of *C. maritimum* (5% *m*/*v*), showed a radical scavenging activity of 87.88%, which corresponds to 7320 mg Trolox/kg of lyophilized plant, assayed by DPPH method.

The supplemented oil samples presented higher radical scavenging activity (16.40–42.66% RSA) than non-supplemented sunflower oil (0.35 ± 0.04 mg Trolox/kg oil). After supplementation, the sunflower oil samples showed the presence of 7.71–29.26 mg Trolox/kg oil. In addition, when the statistical analysis was performed on the antioxidant activity results, no significant effects of lyophilized *C. maritimum* concentration or UAE time on radical scavenging activity were observed. It seems that the increase in the radical scavenging activity of supplemented oils with *C. maritimum* by UAE is explained by other factors rather than the amount of plant that contacted with the oil and/or the UAE time used.

Pereira et al. [7] evaluated the antioxidant activity and the phenolic and mineral contents of *C. maritimum* and found a correlation between the phenolic content and the radical scavenging activity. This finding suggests that the phenolic compounds are the major molecules responsible for the antioxidant capacity of this halophyte. Nevertheless, in our study, no relationship between phenolic content and radical scavenging activity was found.

The results of antioxidant activity of the supplemented sunflower oil samples assayed by FRAP method, are shown in Table 4. Since *C. maritimum* had an antioxidant activity of 20,997 mg Trolox/kg of lyophilized plant and the original sunflower oil only has 2.91 mg Trolox/kg of oil, an increase in the antioxidant activity of sunflower oil was expected after supplementation. As observed in Table 4, the antioxidant activity in supplemented sunflower oil samples varied between 17.50 and 63.62 mg Trolox/kg of oil, which means that the antioxidant activity increased 6 to 22-fold after the extraction of bioactive compounds from *C. maritimum* to the oil, during the ultrasound treatment. A significant linear positive effect of lyophilized plant concentration (*p* = 0.0002) on the antioxidant activity of supplemented oils was observed. The quadratic effect of plant concentration is important enough to be considered in the response surface model. Therefore, the experimental results could be fitted to the response surface in Figure 5 (R^2^ = 0.904; R^2^_adj_ = 0.883) described by the following Equation (6): (6)$$\left[Trolox\right]\left(mg/kg\;oil\right)=16.4826-0.4497\left[plant\right]+0.1505\left[plant^{2}\right]$$
where $\left[plant\right]$ corresponds to the lyophilized *C. maritimum* concentration, in % (*m*/*v*).

The response surfaces fitted to phenolic compounds, flavonoid contents, and antioxidant activity (FRAP assay) have a similar shape. The correlation between antioxidant activity (FRAP assay) and flavonoids (mg quercetin/kg of lyophilized plant) showed a linear dependence, with a determination coefficient of 0.900 (Equation (7)):(7)$$[Trolox]=0.197\left[flavonoids\right]-5.338$$

Similarly, Bakhshi et al. [36], who studied the antioxidant activity of two halophyte plants (*Salsola dendroides* Pall and *Limonium reniforme* Lincz), concluded that the antioxidant activity of the samples was directly related with the content of flavonoid compounds. They also found that the antioxidant activity of the samples was directly related with phenolic content. In the present study, this trend is also observed but no linear relationship was observed. The relationships reported in the literature, between phenolic compounds and antioxidant activity of an oil, are sometimes contradictory because the antioxidant activity depends on the solvent and method of extraction of bioactive compounds, as well as on the phenolic structure and on polymerization degree [37].

From CCRD results, the oil sample with the highest flavonoid and phenolic contents and antioxidant activity was obtained by the extraction of 17.8% (*m*/*v*) of lyophilized *C. maritimum*, for 17.8 min of ultrasound (Table 4). However, the use of more than 12.5% (*m*/*v*) of plant causes a high oil retention in the biomass and, consequently, a hard separation between the supplemented oil and the biomass and low supplemented oil yield. This situation is not viable for the industry. Therefore, the selected conditions for oil supplementation were: 12.5% *m*/*v* of *C. maritimum* and 5 min UAE (experiment 7). The obtained oil corresponds to the sample with the second highest phenolic content and also with the highest radical scavenging activity. Additionally, the amount of lyophilized *C. maritimum* used showed to have a stronger effect than UAE time did in all the assays performed. It means that UAE of bioactive compounds is a very fast process which represents a low energy consumption, which is of utmost importance from an industrial point of view. Thus, the sample obtained with less UAE time was selected for further studies.

### 3.5. Sensory Analysis

The selected supplemented oil sample (12.5% *m*/*v* plant; 5 min UAE) was sensory evaluated to detect smell defects in a scale from 0 (not detected) to 10 (highly perceived) by a profile trained panel. The panelists did not detect any defect or strange smell (rancid: mean and median values equal to 0; other defects: mean and median values equal to 0). Moreover, *C. maritimum* smell was identified but not considered as a defect, with a low-medium level of perception, with 4.5 mean ± 1.3 (minimum value = 2.7; maximum value = 6.2) and 4.9 median.

### 3.6. Accelerated Oxidation Tests

The selected sample (12.5% *m*/*v* plant; 5 min UAE), as well as the original sunflower oil, were submitted to an accelerated oxidation assay for 12 days, in the dark, at 60 °C. For this experiment, a different batch of *C. maritimum,* harvested at a different time (one month later) but in the same location, was used to prepare the supplemented oil samples under the same conditions as experiment 7 of CCRD matrix. This explains the differences between the results on day 0 of this assay and the ones obtained in experiment 7 of the CCRD matrix.

The content of green pigments and carotenoids, and the oxidative stability, assessed by phenolic content and by DPPH and FRAP assays, were evaluated in the samples over the experiment time. Figure 6 presents the evolution of green pigments and carotenoids in the supplemented oil and non-supplemented oil along the storage under accelerated oxidation conditions. The initial content of chlorophyll pigments in the supplemented oil was 26.82 ± 0.56 mg pheophytin a/kg oil which decreased 33% along the experiment following a first-order decay, given by the following Equation (8).
(8)$$\left[Chlorophylls\right]=26.55e^{-0.036t}$$

The concentration of chlorophylls is expressed in mg pheophytin a/kg supplemented oil and the time, t, in day. The value of −0.036 (mg pheophytin a/kg.d) corresponds to the deactivation constant for green pigments oxidation under the storage conditions.

The initial content of carotenoids in the supplemented oil was 24.08 mg β-carotene/kg oil. After 12-d storage at 60 °C, a slight linear decrease of 7.0% was observed, which can be described by the following linear model (Equation (9)):(9)$$\left[Carotenoids\right]=-0.244t\;+24.28$$

Thus, the carotenoids (mg β-carotene/kg oil) in supplemented oil were oxidized at a rate of 0.244 mg/kg.d. The contents of both green pigments and carotenoids in non-supplemented sunflower oil were negligeable.

With respect to the evolution of phenolic compounds, on day 0 of the accelerated oxidation test, supplemented sunflower oil had 47.60 ± 0.183 mg of gallic acid/kg, while non-supplemented sample had 5.43 ± 1.11 mg of gallic acid/kg (almost 8 times less). Along the experiment, the phenolic content of supplemented oil generally decreased, with slight oscillations, reaching 25.84 ± 0.58 mg of gallic acid/kg after 12 days at 60 °C i.e., a 45.7% decrease was observed. Moreover, the phenolic content of non-supplemented sunflower oil sample presented 4.17 ± 0.84 mg of gallic acid/kg at the end of the experiment, which represents around 23% decrease. Although the phenolic content of supplemented oil presented a higher decrease than the non-supplemented oil, the supplemented sunflower oil presented a 6 times higher phenolic content than the non-supplemented sample at the end of experiment period. Similarly, Yang et al. [38] determined the phenolic content of some edible oils supplemented with rosemary extract and non-supplemented oils, during 24 days at 62 °C. Likewise, they also found that despite the decrease, supplemented oils had higher phenolic contents after 24 days at 60 °C. This may be ascribed to the decomposition and oxidation of phenolic compounds, that act like antioxidants and are consumed during the storage period at 60 °C.

Concerning the radical scavenging activity (RSA), the supplemented sunflower oil had a radical scavenging activity of 70.9 ± 6.58% at the beginning of the experiment, which decreased until day five of storage to 50.0 ± 4.88%, at a rate of 3.8%/d (Figure 7). After day 5, a sudden decrease in RSA was observed in supplemented sunflower oil samples, which ended with a RSA of 25.0 ± 2.92%, after 12 days storage. Non-supplemented oil had a RSA of 6.19 ± 0.78% on day 0, and 3.51 ± 0.46% after 12 days under heated storage conditions. Again, the RSA of the supplemented oil was 11 and 7-fold the values observed for the original oil, at the beginning and at the end of the accelerated oxidation test, respectively. Similarly, Siraj et al. [39] studied the oxidative stability of canola and sunflower oil, supplemented with pomegranate seed oil, at 62 °C for 60 days and observed that the radical scavenging activity of the samples decreased during storage time.

In the beginning of the experiment (day 0), the supplemented sunflower oil sample had an antioxidant activity of 52.71 ± 0.17 mg Trolox/kg, which decreased with the storage time and reached 20.02 ± 2.14 mg Trolox/kg after 12 days at 60 °C (62% reduction) (Figure 7). The non-supplemented sunflower oil antioxidant activity remained almost constant over the storage time. Its initial antioxidant activity was 5.61 ± 0.99 mg Trolox/kg and, at the end of the storage, the antioxidant activity of non-supplemented oil was 4.75 ± 2.09 mg Trolox/kg. Again, these results indicate that the supplementation with *C. maritimum* increased the antioxidant activity of sunflower oil about 9-fold and 4-fold, at time 0 and after 12 days storage at 60 °C respectively.

These results are in accordance with the ones obtained by Sousa et al. [14] and Tinello et al. [40]. Sousa et al. [14] monitored the antioxidant activity of sunflower oil enriched with *Pelvetia canaliculata* over storage time (60 °C/12 days), while Tinello et al. [40] evaluated the antioxidant activity of soybean oil supplemented with turmeric and ginger extracts submitted to 62 °C for 28 days. In both cases, a decrease in the antioxidant activity of the oils were observed along storage under accelerated oxidation conditions.

## 4. Conclusions

In this study, bioactive compounds from *C. maritimum* were successfully extracted to sunflower oil by UAE. The supplemented oil samples were prepared following a CCRD matrix, as a function of lyophilized plant concentration and UAE time. The extraction of compounds from *C. maritimum* did not affect the oxidation state of the oil. However, the acidity of the samples was affected by plant concentration and UAE time, but the final acid values were always lower than the legal limit for refined edible oils. The pigment contents (chlorophyll and carotenoids) increased after supplementation, giving a yellow/green colour to the supplemented oils. The addition of the plant did not add smell defects to the oil but gave it a pleasant characteristic smell of *C. maritimum*. 

The flavonoid and phenolic contents, as well as the antioxidant activity, increased in all supplemented samples and a linear correlation between flavonoid content and antioxidant activity was observed.

Regarding both the phenolic content and antioxidant activity of supplemented oil samples, but also considering the oil retention in the lyophilized plant during UAE, the sample obtained with 12.5% (*m*/*v*) of plant, under 5 min of ultrasound, was selected for accelerated oxidation test at 60 °C in the dark. Along this experiment, the phenolic content, the radical scavenging activity, and the antioxidant activity decreased over the test time, but the supplemented samples presented always higher antioxidant properties along the 12-d experiment than the original refined oil. This means that, even under accelerated oxidation conditions, the addition of *C. maritimum*, a halophyte plant of Mediterranean ecosystems, to sunflower oil, increased its phenolic content, as well as RSA and antioxidant activity.

In conclusion, the addition of *Crithmum maritimum* L. clearly add benefits to sunflower oil, increasing its nutritional value and its oxidative stability, and consequently its economic value.

## 5. Patents

The work reported in this manuscript is protected by the following Portuguese patent PT n° 115,610: Tecelão, C.S.R., Neves, M., Silva, S.F.J., Ferreira-Dias, S., Frederico, C.S. Óleos alimentares suplementados com extratos de macroalgas e/ou plantas halófitas. Patente de invenção n° 115,610, INPI, Instituto Nacional de Propriedade Industrial, 25th November 2021, Portugal. [English title: Edible oils supplemented with extracts from macroalgae and/or halophyte plants].

## Figures and Tables

**Figure 1 foods-11-00439-f001:**
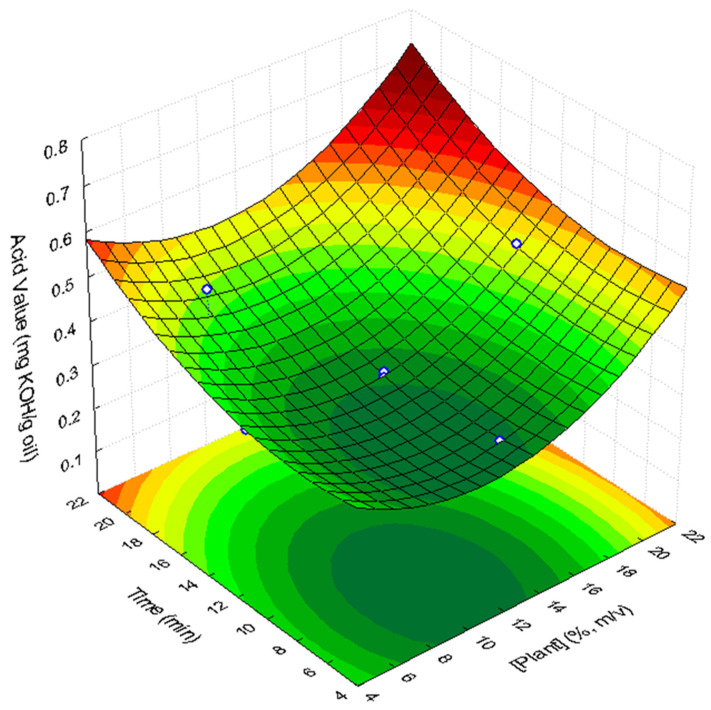
Response surface describing the acid value of supplemented sunflower oil with lyophilized *C. maritimum* extracts, as a function of lyophilized *C. maritimum* concentration and UAE time.

**Figure 2 foods-11-00439-f002:**
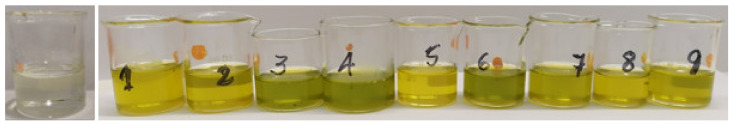
Samples of original (first sample) and supplemented sunflower oil samples with *C. maritimum* extracts by UAE. The extraction conditions of supplemented samples 1 to 9 correspond to those presented in Table 3.

**Figure 3 foods-11-00439-f003:**
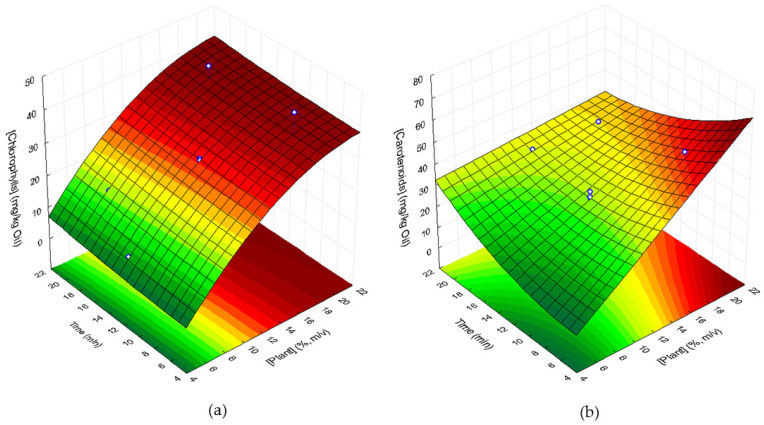
Response surfaces describing the contents of green pigments (chlorophylls), expressed in mg pheophytin a/kg of oil (**a**), and carotenoids, expressed in mg β-carotene/kg (**b**) of supplemented sunflower oil with lyophilized *C. maritimum* extracts, as a function of lyophilized *C. maritimum* concentration and UAE time.

**Figure 4 foods-11-00439-f004:**
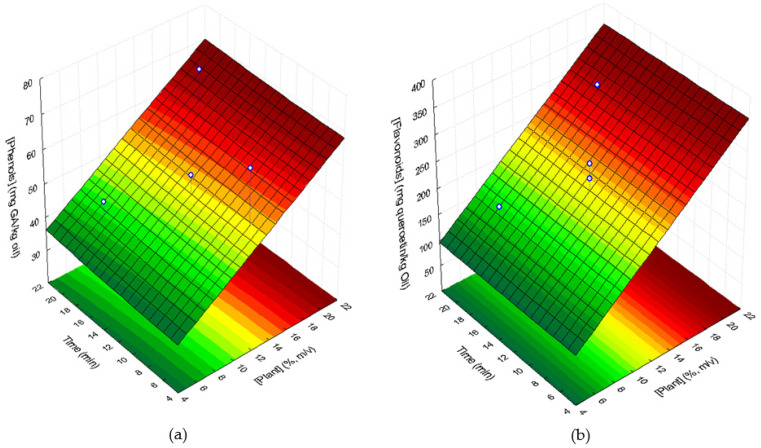
Response surfaces describing the contents of phenolic compounds, expressed in mg gallic acid (GA) equivalent/kg of oil (**a**) and flavonoids, expressed in mg quercetin/kg oil (**b**) of supplemented sunflower oil with lyophilized *C. maritimum* extracts, as a function of lyophilized *C. maritimum* concentration and UAE time.

**Figure 5 foods-11-00439-f005:**
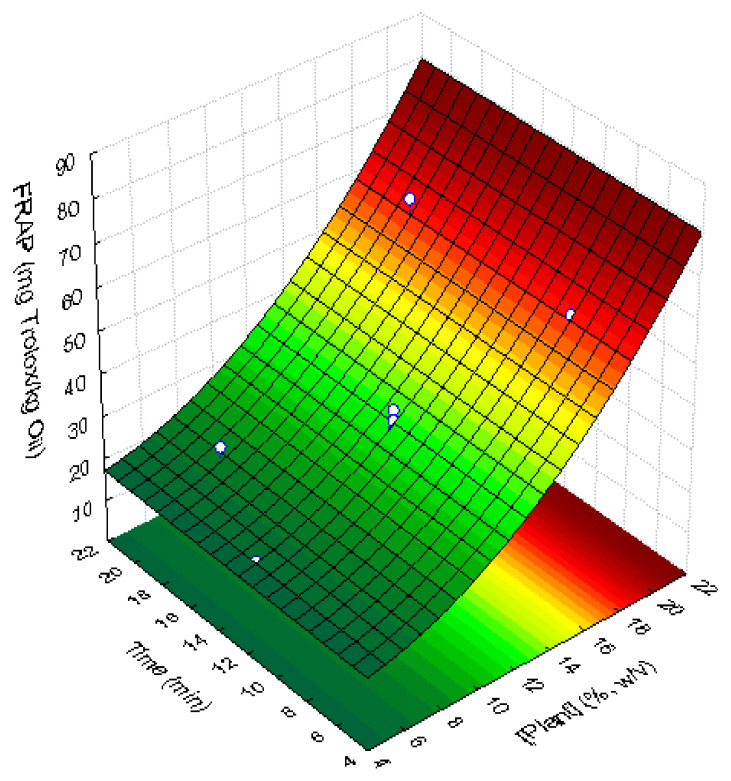
Response surface describing the antioxidant activity, assayed by the FRAP method, expressed in mg Trolox equivalent/kg of supplemented sunflower oil with lyophilized *C. maritimum* extracts, as a function of lyophilized *C. maritimum* concentration and UAE time.

**Figure 6 foods-11-00439-f006:**
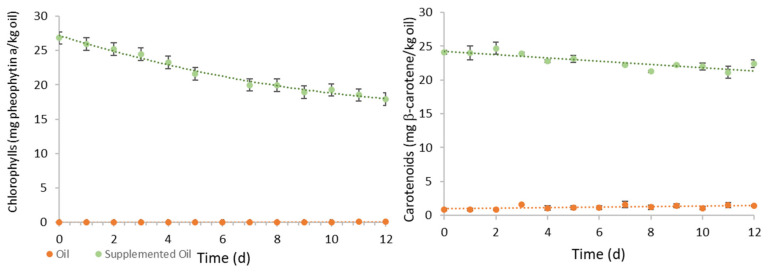
Evolution of the contents of chlorophyll pigments (mg pheophytin a/kg oil) and carotenoids (mg β-carotene/kg oil) in non-supplemented and supplemented oil samples, along storage at 60 °C for 12 days under dark.

**Figure 7 foods-11-00439-f007:**
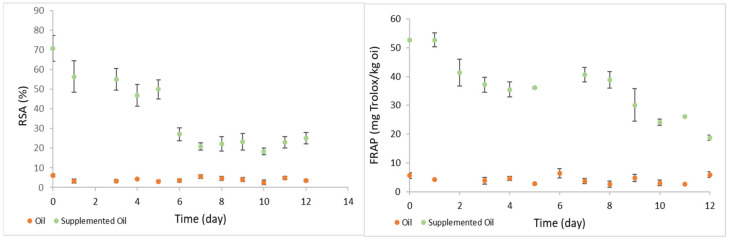
Evolution of the radical scavenging activity (RSA, %) and the antioxidant activity (mg of Trolox equivalent/kg oil), assayed by FRAP method, in non-supplemented and supplemented oil samples, along storage at 60 °C for 12 days under dark.

**Table 1 foods-11-00439-t001:** Levels (values) of the factors (variables) used (UAE time and lyophilized plant concentration) in CCRD experiments for ultrasound-assisted extraction of bioactive compounds from *Crithmum maritimum* to sunflower oil.

Factor (Variable)	Coded Levels				
	$-\sqrt{2}$	−1	0	1	$\sqrt{2}$
(Plant) (% *m*/*v*)	5.0	7.2	12.5	17.8	20.0
UAE Time (min)	5.0	7.2	12.5	17.8	20.0

**Table 2 foods-11-00439-t002:** Concentration of pigments, total phenolic content, and antioxidant activity (assayed by DPPH and FRAP methods) in *Crithmum maritimum* (d.w.) (average values ± standard deviations of three replicates).

Compound	Amount (mg/kg Halophyte)
Chlorophylls	1092 ± 8
Carotenoids (β-carotene)	470 ± 40
Phenolics (gallic acid)	8110 ± 210
Flavonoids (quercetin)	56,202 ± 4019
DPPH, % RSA	87.88 ± 1.66
DPPH (Trolox)	7320 ± 140
FRAP (Trolox)	20,997 ± 222

**Table 3 foods-11-00439-t003:** CCRD results of the green pigments (expressed in mg pheophytin a/kg oil) and carotenoids (expressed in mg β-carotene/kg oil) contents for each sample of supplemented sunflower oil. Sample 13 is the original refined sunflower oil and sample 14 is the original refined oil after a 20 min ultrasound treatment (average values ± standard deviations of three replicates).

Experimental Conditions			Green Pigments		Carotenoids	
Assay	(Plant) (% *m*/*v*) Decoded Values	UAE Time (min) Decoded Values	(mg Pheophytin a/kg Oil)	Extraction Yield (%)	(mg β-Carotene/kg Oil)	Extraction Yield (%)
1	7.2	7.2	14.63 ± 0.06	17.1	19.03 ± 0.62	48.7
2	7.2	17.8	17.83 ± 0.21	20.8	21.79 ± 0.16	56.3
3	17.8	7.2	44.24 ± 0.13	20.9	53.47 ± 0.85	57.9
4	17.8	17.8	42.48 ± 0.08	20.1	42.56 ± 5.15	45.8
5	5.0	12.5	9.14 ± 0.39	15.4	12.63 ± 0.64	44.9
6	20.0	12.5	29.92 ± 0.13	12.6	32.61 ± 1.46	30.9
7	12.5	5.0	27.60 ± 0.20	18.6	33.56 ± 0.63	51.0
8	12.5	20.0	29.16 ± 0.19	19.6	34.66 ± 0.10	52.7
9	12.5	12.5	25.16 ± 0.14	16.9	30.26 ± 0.29	45.8
10	12.5	12.5	29.04 ± 0.15	19.6	32.88 ± 0.36	49.9
11	12.5	12.5	28.56 ± 0.14	19.2	30.52 ± 0.59	46.2
12	12.5	12.5	24.88 ± 0.16	16.7	24.52 ± 0.03	36.7
13	0	0	0.03 ± 0.01	-	1.25 ± 0.66	-
14	0	20	0.02 ± 0.01	-	1.25 ± 0.32	-

**Table 4 foods-11-00439-t004:** CCRD results of phenolic and flavonoid contents, and antioxidant activity of each sample of supplemented sunflower oil. Sample 13 is the original refined sunflower oil (average values ± standard deviations of three replicates). The experimental conditions of the assays are the same as presented in Table 3.

	Phenolic Content		Flavonoids		DPPH		FRAP
Assay	(mg Gallic Acid/kg Oil)	Extraction Yield (%)	(mg Quercetin/kg Oil)	Extraction Yield (%)	% RSA	(mg Trolox/kg Oil)	(mg Trolox/kg Oil)
1	39.05 ± 7.60	5.5	119.39 ± 0.09	1.9	25.52 ± 6.34	12.81 ± 3.18	17.50 ± 1.89
2	47.19	6.8	181.11 ± 16.52	3.4	27.45 ± 0.21	13.72 ± 0.10	26.34 ± 0.39
3	60.24 ± 6.78	3.6	306.76 ± 39.91	2.5	27.90 ± 1.31	21.71 ± 1.02	61.67 ± 22.90
4	72.13 ± 0.67	4.3	319.68 ± 99.56	2.6	35.16 ± 4.97	26.79 ± 3.78	63.62 ± 10.91
5	32.66 ± 9.58	6.5	90.41 ± 2.69	1.9	20.53 ± 2.44	10.29 ± 1.22	18.29 ± 0.39
6	56.38 ± 6.46	3.0	301.11 ±33.09	2.2	30.58 ± 4.88	26.87 ± 4.28	59.99 ± 8.78
7	70.42 ± 7.88	6.0	202.25 ± 20.89	2.2	42.66 ± 4.73	29.26 ± 3.24	36.75 ± 3.61
8	51.33 ± 3.31	4.3	205.54 ± 3.34	2.2	32.16 ± 5.34	23.05 ± 3.82	29.63 ± 7.67
9	56.49 ± 14.55	4.8	-	-	37.92 ± 6.24	29.25 ± 4.80	32.52 ± 2.11
10	44.24 ± 4.50	3.6	244.74 ± 43.15	2.8	26.14 ± 6.90	19.25 ± 5.07	37.52 ± 3.55
11	35.00 ± 1.16	2.8	182.71 ±17.56	2.0	-	-	27.52 ± 1.62
12	45.19 ± 1.74	3.7	272.54 ± 9.03	3.1	16.40 ± 2.02	12.13 ± 1.49	34.93 ± 0.70
13	4.12 ± 0.95	-	33.74 ± 3.01	-	0.64 ± 0.09	0.35 ± 0.04	2.91 ± 0.20

## Data Availability

Data will be available upon request to the authors.

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
