# Peer review of "Enrichment of Sunflower Oil with Ultrasound-Assisted Extracted Bioactive Compounds from Crithmum maritimum L."

_foods, 2022, doi:10.3390/foods11030439_

Round 1

Reviewer 1 Report

File attached. 

Author Response

We kindly thank the reviewer and the editor for the careful revision that clearly improved the manuscript quality.

All the modifications in the manuscript are highlighted in yellow.

Best regards,

Suzana Ferreira-Dias

Reviewer 1

The article is good in terms of signifying importance of Crithmum maritimum L. as an antioxidant for better shelf stability of vegetable oils. Some minor tense mistakes are to be fixed along with general revision of the article.

Some of the sentences and previous studies have been mentioned in present tense, these studies are to be described as past studies for better readability and correlation between previous and current studies. 

I'd suggest to add introductory data from latest articles. A few suggestions are given below:

Shabbir, M. A., Ahmed, W., Latif, S., Inam‐Ur‐Raheem, M., Manzoor, M. F., Khan, M. R., ... & Aadil, R. M. (2020). The quality behavior of ultrasound extracted sunflower oil and structural computation of potato strips appertaining to deep‐frying with thermic variations. Journal of Food Processing and Preservation, 44(10), e14809.

Ordóñez-Santos, L. E., Esparza-Estrada, J., & Vanegas-Mahecha, P. (2021). Ultrasound-assisted extraction of total carotenoids from mandarin epicarp and application as natural colorant in bakery products. Lwt, 139, 110598.

Zardo, I., de Espíndola Sobczyk, A., Marczak, L. D. F., & Sarkis, J. (2019). Optimization of ultrasound assisted extraction of phenolic compounds from sunflower seed cake using response surface methodology. Waste and Biomass Valorization, 10(1), 33-44.

Answer: We do thank the reviewer for the suggestion. However, we decided to reference recently published review articles aiming to give a state-of-the-art perspective on the use of ultrasound assisted extraction in distinct food matrices. The following information was added in the manuscript:

“This technique offers several advantages in comparison with conventional extraction procedures, namely (i) the use of organic solvents is avoided or minimized; (ii) low temperature and extraction time are required, leading to less energy consumption; (iii) high extraction yields are achieved and (iv) the extract quality is preserved [10,11]. UAE has been successfully employed in several food matrices (e.g. fruits, vegetables, and edible oils) for the extraction of bioactive compounds such as polyphenols, carotenoids, and polysaccharides [10–13]” (lines 59-65)

Line 35 – “…hot topic for the food industry”. Rephrase in scientific terms.

Answer: It was rephrased as "… an important issue for the food industry"

Provide brief description on halophyte plants, definition and origin.

Answer: The information was included in the manuscript as follows

“These plants can tolerate extreme environmental conditions, particularly high levels of salinity, enabling its growing in coastal areas and in arid and semiarid regions. In a broad sense, halophytes are recognized as a valuable source of minerals, fibers and some species may also provide important contents of protein and high-quality lipids [4]. Halophytes synthesize various metabolites, namely phenolic compounds that can offer multiple benefits in the food, cosmetics, and pharmaceutical industries due to their antioxidant and antimicrobial activities [4,5]” (lines 38-44).

Specify the wavelength of absorption used for spectrophotometry.

Answer: The information was added in lines 104-105; 111; 115; 126-127; 132.

Corresponded

Answer: It was corrected in line 151.

Provide reference for this methodology.

Answer: The reference for the methodology was added in line 214.

It would be better to specify which method of quantification and environmental conditions/varieties/extraction method were used in previous studies that made such a huge difference in results.

Answer: We thank the reviewer for the suggestion. The information was included in the manuscript as follows:

“Renna et al. [28] determined the chlorophyll content of C. maritimum collected in Italy, using an extraction procedure with 80% acetone followed by spectrophotometric quantification at 647 and 664 nm. Values of 32.71 ± 3.59 mg for chlorophyll a and 9.75 ± 0.86 mg for chlorophyll b per 100 g of freeze-dried halophyte were attained.” (lines 247-250)

“Nabet et al. [29] determined the content of carotenoids of C. maritimum harvested in Algeria by carrying out a double extraction with n-hexane/acetone/ethanol (2/1/1, v/v/v). After washing the mixture with distilled water, the n-hexane phase was recovered for spectrophotometric quantification of total carotenoids at 450 nm. The results obtained by these authors (62.6 ± 3.8 mg of β-carotene/kg of dry weight) were almost 8 times smaller than the results obtained in the present study.” (lines 253-259)

This phrase is repeated excessively (“These differences may be explained by different environ-255 mental conditions to which the plants were exposed, as well as the use of a higher extrac-256 tion time that probably allowed a more effective extraction”). Only use it at the end of after elaborating all the studies relating to a specific analysis.

Answer: We agree with the reviewer. The text was rewritten as follows

“These differences may be explained by several factors that influence the phenolic content of halophyte plants, namely the harvest time, biotic/abiotic stressors (e.g. salinity, UV radiation, extreme temperatures and exposure to pollutants), growth conditions (water and soil composition) and extraction methodology [4]”. (lines 277-280)

Sensory analysis

Provide these results in form of graph. Results of all nine panelists/assessors should be included.

Answer: As you know, the results from a sensory panel are always given by the mean value of individual scores or, in the case of virgin olive oil, by the median value, according to the European legislation (COMMISSION REGULATION (EEC) No 2568/91). Each individual response acts as a repetition and, as for any other laboratory analysis, we never present all the repetitions but only the mean value and standard deviation, or in this case, the median value for each attribute. Therefore, we added the standard deviation, the minimum and maximum values to the mean values of each attribute, as follows:

The panelists did not detect any defect or strange smell (rancid: mean and median values equal to 0; other defects: mean and median values equal to 0). Moreover, C. maritimum smell was identified but not considered as a defect, with a low-medium level of perception, with 4.5 mean ± 1.3 (minimum value= 2.7; maximum value= 6.2) and 4.9 median.

Reviewer 2 Report

Enrichment of Sunflower Oil with Ultrasound-Assisted Ex-2 tracted Bioactive Compounds from Crithmum maritimum L.

Lines 12 and 47. Change the word “spontaneously” to another more suitable word, for example “naturally”.

Keywords: They suggested not repeating the words of the title in the keywords.

Line 50. Are all the references that report these properties?

Line 52-61. The paragraph is confusing. Reorganize the sentence since the objective is not clear. There are also sentences like a methodology and a justification.

Lines 65-37. Please clarify if this plant was correctly identified by a recognized herbarium. If so, please add the registration number.

Lines 74-75. Specify which part of the plant was used (leaves, stems, whole plant). Also specify the amount of biomass (in grams) used for the extraction process.

77-78. Specify how many times the extraction process was done. Was it only once?

77-78. The authors must justify why they used an unusual technique for the extraction process. The most common is to reflux for 20 minutes and extract 3 times on the same sample. If it is required to avoid high temperature, it is better to do the extraction process by maceration for at least 12 hours and it should be repeated at least 3 times. Why was it not done this way? It is strange that it was only 6 minutes using a vortex, this is not adequate because a good extraction of most of the components is not ensured.

Line 103 and 108. Specify all concentrations used to run the calibration curve. In my opinion 0.5 mg/mL or 1.2 mg/mL (minimum concentration used) are such high concentrations compared to other studies. Please review those data.

The authors must attach as complementary files the calibration curves for each of the experiments carried out.

Table 2. Restructure the Table, mg/mL is so repetitive. This can be added in the second column.

The conclusions are so long, please shortened it.

Review the Instructions for Authors for References section.

Author Response

We kindly thank the reviewer and the editor for the careful revision that clearly improved the manuscript quality.

All the modifications in the manuscript are highlighted in yellow.

Best regards,

Suzana Ferreira-Dias

Reviewer 2

Enrichment of Sunflower Oil with Ultrasound-Assisted Extracted Bioactive Compounds from Crithmum maritimum L.

Lines 12 and 47. Change the word “spontaneously” to another more suitable word, for example “naturally”.

Answer: done

 Keywords: They suggested not repeating the words of the title in the keywords.

Answer: we removed those that were in the title (e.g bioactive compounds; Crithmum maritimum L.; sunflower oil) and added “pigments, phenolic compounds, flavonoids”.

Line 50. Are all the references that report these properties?

Answer: this part of the text was modified and completed with references, as suggested by reviewer 1.

Line 52-61. The paragraph is confusing. Reorganize the sentence since the objective is not clear. There are also sentences like a methodology and a justification.

Answer: This paragraph was modified as follows:

“In this sense, the purpose of this study was to supplement sunflower oil with bioactive compounds from C. maritimum L., directly extracted from the plant to the oil by UAE, aiming at increasing the nutritional and economic values and oxidative stability of the oil. UAE process was performed under the conditions dictated by a central composite rotatable design (CCRD) as a function of lyophilized plant/oil ratio and ultrasound application time, to find the optimized conditions. The storage stability of sunflower oil enriched with bioactive compounds from C. maritimum, under selected UAE conditions, was additionally assessed at 60 °C.”

Lines 65-37. Please clarify if this plant was correctly identified by a recognized herbarium. If so, please add the registration number.

Answer: This plant was recognized by a colleague with strong knowledge of botanic. This plant is very common in Portuguese shore and there is no danger to confuse it with another one. Unfortunately, we cannot add the registration number.

Lines 74-75. Specify which part of the plant was used (leaves, stems, whole plant). Also specify the amount of biomass (in grams) used for the extraction process.

Answer: The whole plant was used. This information was added.

77-78. Specify how many times the extraction process was done. Was it only once?

Answer: It was only one time.

77-78. The authors must justify why they used an unusual technique for the extraction process. The most common is to reflux for 20 minutes and extract 3 times on the same sample. If it is required to avoid high temperature, it is better to do the extraction process by maceration for at least 12 hours and it should be repeated at least 3 times. Why was it not done this way? It is strange that it was only 6 minutes using a vortex, this is not adequate because a good extraction of most of the components is not ensured.

Answer: Thank you very much for your suggestions. You are right. We decided to use this methodology for bioactive extraction from the plant, to be able to compare with a previous work on UAE of bioactive compounds from the seaweed Pelvetia canaliculata to sunflower oil, where the same methodology was used.

Line 103 and 108: Specify all concentrations used to run the calibration curve. In my opinion 0.5 mg/mL or 1.2 mg/mL (minimum concentration used) are such high concentrations compared to other studies. Please review those data.

Answer: These values are the maximum values and not the minimum (concentrations up to). In the first version it is written:

Line 103: in concentrations up to 0.5 mg/mL.

Line 108: “in concentrations up to 1.2 mg/mL”

The authors must attach as complementary files the calibration curves for each of the experiments carried out.

Answer: Done.

Table 2. Restructure the Table, mg/mL is so repetitive. This can be added in the second column.

Answer: We deleted mg/kg halophyte from the column 1 and put is in the heading (Amount (mg/kg halophyte)

The conclusions are so long, please shortened it.

Answer: we apologize but, since none of the other two reviewers requested to shorten this section, we decided to maintain it as in the first version of the manuscript.

Review the Instructions for Authors for References section.

Answer: done

Reviewer 3 Report

In the manuscript titled " Enrichment of Sunflower Oil with Ultrasound-Assisted Extracted Bioactive Compounds from Crithmum maritimum L.", the authors study the enrichment of sunflower oil with bioactive compounds from sea fennel. The paper is interesting, well written and should go on major revision according to the comments listed:

Introduction: some more information about the application of the UAE method for direct oil fortification should be included in the manuscript. Also, the introduction should include some examples of the use of DOE (experimental design) for UAE optimization.

The novelty of the research should also be highlighted in the introduction.

Line 72. The lyophilization conditions should be listed.

Line 77. What was the volume of the extraction mixture?

Line 78. What was the mixing rate?

Section 2.2.1., 2.2.2., 2.2.3. were the analyses performed with replicates?

Section 2.4.1., 2.4.2., 2.4.3., 2.4.4. were the analyses performed with replicates?

Section 2.7. provides a description of RSM modelling only. What about other methods used for descriptive statistical analysis of the data?

Line 238. what quantification method was used by Nabet et al? Please explain in more detail.

Line 283 and Equation 1. The authors state that "statistical analysis showed that both VAE time and plant loading had a positive linear effect on acidity (p= 0.03 and p= 0.10, respectively)", but in Equation 1 there is a negative sign in front of the linear coefficient for plant loading. Please explain.

Line 312. it would be interesting to analyze the total color change data (△E) with respect to the original sunflower oil.

Equation 2 should be included in the Materials and Methods section, as well as the explanation of this equation.

Please use the dot as the multiplication sign in all equations in the manuscript.

Table 3. and Table 4 Extraction yields should also be reported with standard deviations.

Line 498. how were the optimal extraction conditions determined? The authors should present the results of the independent validation experiment for the optimal extraction conditions.

For the data presented in Figures 6 and 7, it would be interesting to calculate the degradation rate to accurately define the oxidation effects during storage.

Author Response

We kindly thank the reviewer and the editor for the careful revision that clearly improved the manuscript quality.

All the modifications in the manuscript are highlighted in yellow.

Best regards,

Suzana Ferreira-Dias

Reviewer 3

In the manuscript titled "Enrichment of Sunflower Oil with Ultrasound-Assisted Extracted Bioactive Compounds from Crithmum maritimum L.", the authors study the enrichment of sunflower oil with bioactive compounds from sea fennel. The paper is interesting, well written and should go on major revision according to the comments listed:

Introduction: some more information about the application of the UAE method for direct oil fortification should be included in the manuscript. Also, the introduction should include some examples of the use of DOE (experimental design) for UAE optimization.

The novelty of the research should also be highlighted in the introduction.

Answer: The introduction section was imporved.

Line 72. The lyophilization conditions should be listed.

Answer: The required information was added in the manuscript as follows:

“The halophyte was lyophilized at – 56°C, for 3 days under vacuum (freeze-dryer Telstar, Lyoquest-85, Telstar Portugal, Lisbon, Portugal)” (lines 81-82)

Line 77. What was the volume of the extraction mixture?

Answer: We apologize but we do not understand your question. Since we inform the readers that “Methanolic and n-hexane extracts of the lyophilized plant (5% m/v) were prepared by vortex stirring the biomass in the solvent for 6 min”, the proportion lyophilized plant/solvent is indicated (5 %, m/v).

Line 78. What was the mixing rate?

Answer: We used a current laboratory vortex stirrer with manual motor activation. We do not know the attained mixing rate.

Section 2.2.1., 2.2.2., 2.2.3. were the analyses performed with replicates?

Answer: All analyses were performed in triplicate. This information was added to the manuscript at the end of these sections.

Section 2.4.1., 2.4.2., 2.4.3., 2.4.4. were the analyses performed with replicates?

Answer: Also, three replicates of each sample were analyzed. This information was added to the manuscript at the end of these sections.

Section 2.7. provides a description of RSM modelling only. What about other methods used for descriptive statistical analysis of the data?

Answer: In this study, we performed the optimization of ultrasound-assisted extraction of bioactive compounds from lyophilized C. maritimum to sunflower oil, using RSM. As you can see along the manuscript, the descriptive analysis of the data was only the calculation of mean values and respective standard deviations for each analytical determination. Since RSM was used, there was no need to perform ANOVA to evaluate if each result from the CCRD was significantly different from the others.

Concerning the fit of the first-order decay model to the results of the decay of chlorophyll pigments content along accelerated oxidation test, the MS Excel “Solver” supplement was used. The fit was performed by minimising the residual sum-of squares between the experimental data points and those estimated by the respective model.

This information was added to the Section 2.7 and the model fitted to the experimental data was added to Section 3.6 Accelerated oxidation tests.

Line 238. what quantification method was used by Nabet et al? Please explain in more detail.

Answer: The methodology was explained in more detail, as follows:

“Nabet et al. [29] determined the content of carotenoids of C. maritimum harvested in Algeria by carrying out a double extraction with n-hexane/acetone/ethanol (2/1/1, v/v/v). After washing the mixture with distilled water, the n-hexane phase was recovered for spectrophotometric quantification of total carotenoids at 450 nm. The results obtained by these authors (62.6 ± 3.8 mg of β-carotene/kg of dry weight) were almost 8 times smaller than the results obtained in the present study.” (lines 253-259)

Line 283 and Equation 1. The authors state that "statistical analysis showed that both VAE time and plant loading had a positive linear effect on acidity (p= 0.03 and p= 0.10, respectively)", but in Equation 1 there is a negative sign in front of the linear coefficient for plant loading. Please explain.

Answer: The coefficients of the model equations describing the response surface fitted to the experimental data are not the effects. They are directly related with the effects only when the model equation is given in coded values (-alpha, -1, 0, +1, + alpha), which is not the case.

We decided not to include the tables of the effects in the manuscript since it is not a statistical manuscript and it would difficult the reading for most of the readers which may be not very familiar with Experimental Design and Response Surface Methodology.

Please see below the effects (linear and quadratic terms) for each factor considered in the model fitted to Acid value results:

DV: Acid Value

Effect

Std.Err.

t(7)

p

Mean/Interc.

0.33

0.02

14.51

0.00

(1)[Plant](L)

0.06

0.03

2.01

0.08

[Plant](Q)

0.10

0.04

2.84

0.03

(2)Time (min)(L)

0.10

0.03

3.08

0.02

Time (min)(Q)

0.05

0.04

1.48

0.18

Line 312. it would be interesting to analyze the total color change data (△E) with respect to the original sunflower oil.

Answer: Thank you for the suggestion but now we do not have supplemented oils to perform the analysis and it was not possible to repeat the experiments in the time given for the revision of the manuscript. We will keep in mind your suggestion for further studies.

Equation 2 should be included in the Materials and Methods section, as well as the explanation of this equation.

Answer: We apologize but we do not agree with your suggestion: equation 2 is a result of the statistical treatment of the results obtained for chlorophyll contents in the supplemented oils, as a function of UAE time and plant concentration. It must be in the section of Results and not in the section of Materials and Methods

Please use the dot as the multiplication sign in all equations in the manuscript.

Answer: the equations were written using the MS Word equation writer where multiplication is given by “X”. Replacing it by a dot will be very strange because we will also have a dot to separate the units from the decimals. Therefore, we decided to remove “X”, but do not add a dot.

Table 3. and Table 4 Extraction yields should also be reported with standard deviations.

Answer: The extraction yields were calculated using the average values and not the individual values. Therefore, they do not present standard deviations.

Line 498. how were the optimal extraction conditions determined? The authors should present the results of the independent validation experiment for the optimal extraction conditions.

Answer: The experiments were carried out under the extraction conditions indicated in the manuscript: 12.5 % m/v of C. maritimum and 5 min UAE (experiment 7). The obtained results for each assayed compound or property correspond to the initial values (time 0) shown in figures 6 and 7 and also referred in the manuscript (section 3.6 Accelerated oxidation tests).

For the data presented in Figures 6 and 7, it would be interesting to calculate the degradation rate to accurately define the oxidation effects during storage.

Answer: For chlorophyll pigments (Fig. 6), the fit of first-order decay allows to estimate the oxidation effects on these compounds. For carotenoids, a linear decrease was observed, and the slope gives us the degradation rate. This information was added to the manuscript, as follows:

“The initial content of chlorophyll pigments in the supplemented oil was 26.82±0.56 mg pheophytin a/kg oil which decreased 33 % along the experiment following a first-order decay, given by the following equation (Eq. 8).

The concentration of chlorophylls is expressed in mg pheophytin a/kg supplemented oil and the time, t, in day. The value of -0.036 (mg pheophytin a/kg.d) corresponds to the deactivation constant for green pigments oxidation under the storage conditions.

The initial content of carotenoids in the supplemented oil was 24.08 mg b-carotene/kg oil. After 12-d storage at 60 ℃, a slight linear decrease of 7.0 % was observed, which can be described by the following linear model (Eq. 9):

 Thus, the carotenoids (mg β-carotene/kg oil) in supplemented oil were oxidized at a rate of 0.244 mg/kg.d.”

Concerning Fig. 7 (radical scavenging activity and the antioxidant activity), it is not possible to fit a model to the values decay and, therefore, no degradation rate can be accurately estimated from these data.

Round 2

Reviewer 1 Report

Great revision. 

Reviewer 3 Report

The authors put an effort and answered the comments and suggestions. In my opinion, they have improved the manuscript and therefore I think that the manuscript can be accepted for publication.